# Dual-Branch HNSW Approach with Skip Bridges and LID-Driven Optimization

## Abstract

The Hierarchical Navigable Small World (HNSW) algorithm is widely used for approximate nearest neighbor (ANN) search, leveraging the principles of navigable small-world graphs. However, it faces some limitations. The first is the local optima problem, which arises from the algorithm's greedy search strategy, selecting neighbors based solely on proximity at each step. This often leads to cluster disconnections. The second limitation is that HNSW frequently fails to achieve logarithmic complexity, particularly in high-dimensional datasets, due to the exhaustive traversal through each layer. To address these limitations, we propose a novel algorithm that mitigates local optima and cluster disconnections while enhancing the construction speed, maintaining inference speed. The first component is a dual-branch HNSW structure with LID-based insertion mechanisms, enabling traversal from multiple directions. This improves outlier node capture, enhances cluster connectivity, accelerates construction speed and reduces the risk of local minima. The second component incorporates a bridge-building technique that bypasses redundant intermediate layers, maintaining inference and making up the additional computational overhead introduced by the dual-branch structure. Experiments on various benchmarks and datasets showed that our algorithm outperforms the original HNSW in both accuracy and speed. We evaluated six datasets across Computer Vision (CV), and Natural Language Processing (NLP), showing recall improvements of 18% in NLP, and up to 30% in CV tasks while reducing the construction time by up to 20% and maintaining the inference speed. We did not observe any trade-offs in our algorithm. Ablation studies revealed that LID-based insertion had the greatest impact on performance, followed by the dual-branch structure and bridge-building components.

## 1 Introduction

Hierarchical Navigable Small World (HNSW) graphs have become a state-of-the-art method for approximate nearest neighbor (ANN) search due to their efficiency and effectiveness in handling large-scale datasets (Malkov & Yashunin, 2020). HNSW constructs a multi-layer graph, where each layer provides a different level of abstraction of the data. The search process navigates these layers, starting from the top, to efficiently approximate the nearest neighbors of a query point.

Despite its success, HNSW faces several limitations. The first drawback relates to local optimum during the search process. This issue arises from the node insertion mechanism, which inserts nodes into the HNSW graph randomly. Random insertion can result in disconnected regions and weaker inter-cluster connectivity, increasing the likelihood of the search process becoming trapped in local optimum. As shown in Figure 1, random insertion in the HNSW graph makes the greedy search process traverse to a local optimum (denoted by the red node), instead of reaching the global optimum (denoted by the green node). The second drawback is that the logarithmic complexity $\mathcal{O}(n \log n)$ proposed in the original HNSW paper (Malkov & Yashunin, 2020) is difficult to achieve consistently, particularly in high-dimensional spaces (Lin & Zhao, 2019). The exhaustive traversal through each layer introduces significant overhead, slowing both the construction and query processes.

To address the aforementioned limitations, we propose a novel approach to constructing HNSW graphs by inserting nodes based on their Local Intrinsic Dimensionality (LID) values instead of

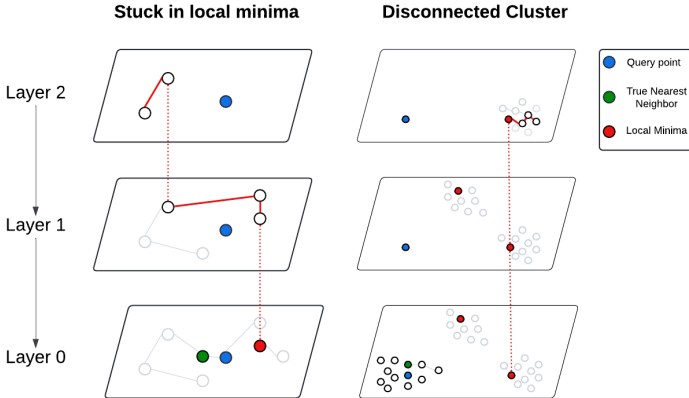

Figure 1: Illustration of local minima and disconnected regions problems in Hierarchical Navigable Small World (HNSW): the algorithm frequently gets trapped in local minima and suffers from inadequate inter-cluster connectivity

using random insertion, thereby overcoming the high risk of being stuck in local optima. By prioritizing nodes with higher LID values, we aim to better capture outlier nodes within clusters, thereby enhancing inter-cluster connectivity. Additionally, we propose using LID thresholds to effectively build bridges that skip layers, which accelerates the query process and makes achieving $\mathcal{O}(n \log n)$ complexity more feasible. Furthermore, we introduce a dual-branch structure in the HNSW graph to further mitigate the risk of local minima and to speed up the construction time.

Our main contributions are as follows:

- A dual-branch HNSW graph structure that increases the diversity of navigation paths, reduces construction time, and mitigates search stagnation in local optima, thereby improving recall and accuracy.
- A LID-based node insertion strategy for HNSW graphs, prioritizing outliers to higher layers to improve inter-cluster connectivity, further reducing local optima and enhancing performance.
- LID-based threshold layer-skipping bridges that significantly accelerate the query process.
- Through experiments, we demonstrate that our approach enhances search accuracy and construction time while maintaining inference time. We do not observe any trade-offs in our algorithm.

## 2 RELATED WORK

### 2.1 HIERARCHICAL NAVIGABLE SMALL WORLD

HNSW graphs are prominent due to their efficiency and scalability across various datasets. (Malkov & Yashunin, 2020) introduced HNSW as a multi-layered graph structure that allows efficient neighborhood exploration in high-dimensional spaces. The performance of HNSW is highly dependent on its parameters, such as the number of links per node ($M$) and search complexity ($ef\_Search$), but its success also comes from the log distribution structure of the data being indexed.

Recent advancements have significantly deepened the understanding and extended the capabilities of HNSW, driving state-of-the-art improvements in its performance and applications. (Zhang et al., 2022)) explored optimization techniques tailored for large-scale HNSW graphs, focusing on enhancing search efficiency through refined graph construction. Cole Foster (2023) further advanced these efforts by combining hybrid graph-based approaches with HNSW enhancements, pushing the boundaries of accuracy in ANN search. Complementing these innovations, (Elliott & Clark, 2024) highlighted the critical influence of data insertion order on intrinsic dimensionality and overall recall performance. Together, these studies paint a cohesive picture of the evolving landscape of

HNSW research, showcasing a continuous push toward more efficient and effective graph-based search methodologies.

## 2.2 High Local Intrinsic Dimensionality in HNSW

One of the key challenges in Approximate Nearest Neighbor (ANN) search is dealing with local minima, where the search process gets stuck in suboptimal regions of the data space. High Local Intrinsic Dimensionality (LID) values, as demonstrated by (Houle et al., 2018), indicates its role in capturing the density of data points around the query, serving as a measure of local intrinsic dimensionality. High-LID values often represent outliers or points in sparse, locally complex regions, which can disrupt the search process by anchoring the graph structure in unfavorable ways, mitigating the risk of local minima.

The importance of handling high-LID outliers in graph construction was also explored by (Amsaleg et al., 2015), who proposed several methods for estimating LID and demonstrated its usefulness in ANN search, classification, and outlier detection. Their findings support the idea that high-LID points should be treated differently during graph construction.

Recent work by (Elliott & Clark, 2024) has highlighted the significant impact of data insertion order on the recall performance of HNSW graphs. Their study showed that inserting nodes with higher LID values into the upper layers of the HNSW graph can improve recall by up to 12.8 percentage points.

While these works offer valuable insights into handling high-dimensional data and mitigating local minima, they leave open questions regarding the best methods for integrating LID considerations into HNSW graph construction. Furthermore, the impact of LID-based strategies on the broader structure of HNSW graphs remains an area for further exploration.

## 3 HNSW++: Dual-branch HNSW approach with bridges and LID-Driven Optimization

### 3.1 Motivation

In HNSW, nodes are assigned to layers randomly, with the probability of a node being placed in a higher layer decreasing exponentially. In (Baranchuk et al., 2019), the authors pointed out that similarity graphs are vulnerable to local minima when the query is unable to escape suboptimal vertices. The combination of random insertion and greedy search in HNSW worsens this issue, as random factors can cause the search to deviate from the optimal path. Another challenge arising from the random insertion is the low inter-cluster connectivity. This problem can place different clusters on separate layers, limiting their connectivity. Consequently, the search process may terminate in a cluster different from the one containing the true nearest neighbors. (Lin & Zhao, 2019) also observed that the hierarchical structure of HNSW fails to achieve the expected logarithmic complexity. Instead, the exhaustive traversal of each layer becomes a bottleneck. As a result, HNSW encounters several inherent problems: (1) a high likelihood of local minima, which grows with the size of the data; (2) weak connectivity between clusters; and (3) slower search times, construction time, making it difficult to achieve logarithmic complexity in practice.

### 3.2 Proposed Methodology

To address these issues, we propose HNSW++, which partitions the dataset into two branches based on the index of the inserted nodes. By doing so, spatial regions are divided into different branches, allowing the algorithm to search in both simultaneously. The search process begins in the upper layers of both branches, navigating greedily until a local minimum is encountered. The algorithm then descends to the lower layers, where the two branches merge at layer 0, combining their results for the final output (see Fig. 3). This approach not only minimizes the influence of local minima and ensures a more accurate search for the true nearest neighbors, but it also reduces construction time, as each new node only needs to search through half the nodes already inserted. This dual-branch approach can be expressed as follows:

$$\text{HNSW++}(D) = \text{Merge}\left(S(q, L_1, exclude\_set_1), S(q, L_2, exclude\_set_2), k\right) \qquad (1)$$

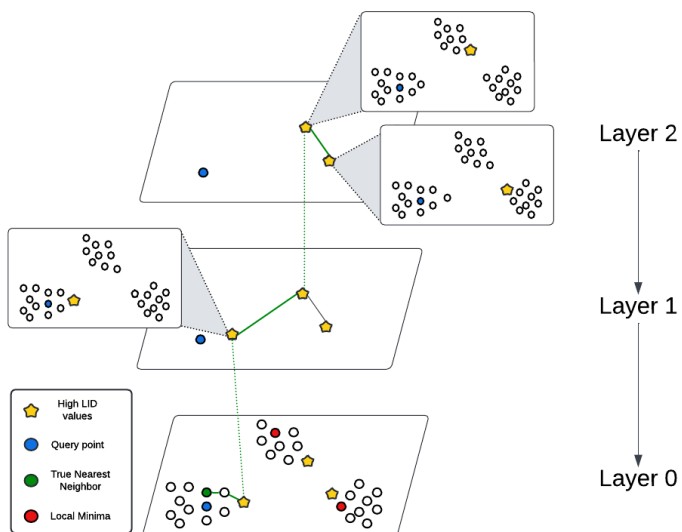

Figure 2: Illustration of the method for inserting high-LID nodes into the top layers

where $D$ is the dataset containing $n$ nodes, $q$ is the query node. $S(x, l, exclude\_set)$ is the search process for the nearest neighbor of node $x$, which begins at layer $l$, and $exclude\_set$ is the result from other branch if available. The $exclude\_set$ is passed to the Search algorithm in layer 0 of both branches to ensure that the neighbors returned by each branch are distinct, preventing overlap in the results between the two branches. $L_1$ and $L_2$ are the upper layers of branch 1 and branch 2, respectively. Among the neighbors returned by $S(x_q, L_1, exclude\_set_1)$ and $S(q, L_2, exclude\_set_2)$, Merge function selects the $k$ neighbors that are closest to the query node $q$.

Additionally, to improve cluster connectivity, we utilize Local Intrinsic Dimensionality (LID) values during insertion into HNSW++. The Local Intrinsic Dimensionality (LID) of a data point can be estimated using Maximum Likelihood Estimation (MLE) combined with the $k$-Nearest Neighbors ($k$NN) algorithm (Hand et al., 2001). The Maximum Likelihood Estimate of the LID, $LID(x)$, is given by the following formula:

$$LID(x) = \left( \frac{1}{k-1} \sum_{i=1}^{k-1} \log \frac{d_k}{d_i} \right)^{-1} \tag{2}$$

where $d_i$ is the Euclidean distance between the query point and its $i$-th nearest neighbor, while $d_k$ is the distance to the $k$-th nearest neighbor (Levina & Bickel, 2004). High LID scores indicate sparse regions, often found at the edges of clusters (see Fig. 2). By inserting nodes with high LID into upper layers, we facilitate connections between clusters. During the search, these nodes enable faster traversal between clusters in the upper layers, and a more precise search within the target cluster in the lower layers. This strategy reduces the chances of the search becoming stuck in a suboptimal cluster, guiding it more efficiently toward the true nearest cluster. This LID estimation plays a crucial role in guiding the construction and optimization of HNSW graphs by allowing us to distinguish between nodes that exist in regions of varying density, thus improving the efficiency and accuracy of search operations. This expanded version provides more context about LID and its importance while still preserving the meaning and structure of your original section.

To further accelerate the search and approximate logarithmic complexity in practice, we introduce a method for creating additional skip bridges between upper layers and bottom layer (layer 0) based on LID thresholds (see Fig. 3). During the search, as nodes are traversed, their corresponding LID values are evaluated. If a node's LID exceeds the threshold, indicating a sparse distribution of nodes between the considered node and the query node, and the distance between the two nodes is near enough, the search can directly jump to layer 0. This approach bypasses intermediate layers, minimizing the need for exhaustive traversal across layers. This mechanism captures both the distance

from the current node to the query and the sparsity of the surrounding area, enabling the creation of shortcuts in the graph, ultimately speeding up the query process. The mathematical expression of skip bridges can be seen in Equation 3:

$$S_{\text{skip}}(q, L_l) = \begin{cases} S(q, 0, exclude\_set) & \text{if Jump}(ep, q) \text{ is True} \\ S(q, L_l) & \text{for layers } L_l \text{ otherwise} \end{cases} \quad (3)$$

where $S_{\text{skip}}$ denotes the search process utilizing skip bridge to search for nearest neighbor of query node $q$. $exclude\_set$ is the result from other branch if available and is compulsory if $L_l$ is 0. Jump$(ep, q)$ is a boolean function that determines whether the search can jump directly to layer 0 based on the LID and distance conditions between the current node $x_i$ and the query node $x_q$. The equation for Jump$(x_i, x_q)$ is determined as below:

$$\text{Jump}(ep, q) = \begin{cases} \text{True} & \text{if LID}(ep) > T \text{ and } d(ep, q) < \epsilon \\ \text{False} & \text{otherwise} \end{cases} \quad (4)$$

where $ep$ is the entry point of current layer which is also the nearest node from $W$ of previous layer , $q$ is the query node, LID$(ep)$ is the LID value for $ep$, $T$ is the LID threshold value, $d(ep, q)$ is the distance metric between $ep$ and query node $q$, $\epsilon$ is a distance threshold that indicates when two nodes are considered "near enough".

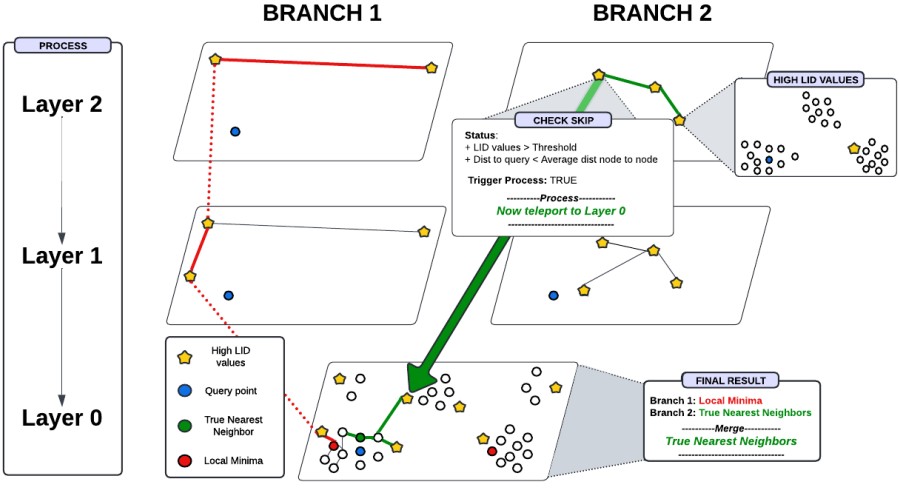

Figure 3: Illustration of the method for inserting high-LID nodes into the top layers, 2 branches and the skip-layer method based on a threshold.

## 3.3 ALGORITHM DESIGN

The network construction algorithm incrementally inserts nodes into the hierarchical structure based on their assigned layer and order. Starting at the top layer, the algorithm searches for the closest nodes to the query, then progresses to the next lower layer, using the previous layer's result as the entry point. This continues iteratively until reaching the query node's designated layer. From there, connections between the query node $q$ and surrounding nodes are established, with further refinement through adding or dropping connections as limits are exceeded.

Given an array of original LID values, the layer assignment for new nodes is guided by their normalized Local Intrinsic Dimensionality (LID), as detailed in Algorithm 4. Nodes are divided between two branches, branch$_0$ and branch$_1$, ranked by descending normalized LID calculated by Algorithm 5. The number of nodes per layer is determined by a random scaling factor based on a normalization constant $m_L$, ensuring a balanced distribution. If the total node count is odd, branch$_0$

receives an extra node. The algorithm alternates between branches during assignment, and once a branch reaches capacity, remaining nodes are assigned to the other branch.

In the construction phase (Algorithm 1), the insertion process alternates between the two branches to ensure an even node distribution, with each branch handling approximately half of the dataset.

For inference (Algorithm 2), a modified greedy search algorithm locates the nearest neighbors for query node $q$, starting from the top layer. Separate search paths are initialized for each branch, with $W_1$ and $W_2$ as entry points. Both branches search concurrently at layer 0 (Algorithm **??**). The inference phase also employs a layer-skipping mechanism—if the distance between the current node and the query is below the dataset's average distance and has a high normalized LID, the search can directly skip to layer 0, bypassing intermediate layers. This reduces the search time, while the other branch continues unless also skipped.

When one branch finishes first, it passes its nearest neighbor set to the other branch to avoid duplicate results. If both branches finish simultaneously, one branch's results are passed to the other to ensure consistency (Algorithm **??**). Finally, the neighbor sets from $W_1$ and $W_2$ are combined. This dual-path approach mitigates local minima by exploring two distinct search regions.

### 3.4 LID THRESHOLD INFLUENCE ON NUMBER OF SKIPS AND PERFORMANCE

During the inference stage, the LID threshold determines whether a search can directly jump to layer 0, bypassing redundant intermediate layers, thereby reducing inference time (Algorithm 2). Normalized LID values range from 0 to 1, where a value of 1 indicates that the neighborhood around a specific node is sparse, while a value of 0 reflects a dense neighborhood.

To achieve optimal performance, multiple experiments on LID threshold were conducted, including (1) Influence of threshold on number of skips, (2) Influence of threshold on accuracy and recall.

The impact of the LID threshold on the number of skips is illustrated in Figure 14, clearly showing that as the threshold increases, the number of skips decreases across all datasets. At lower thresholds, the algorithm performs more frequent skips, enhancing search efficiency in the denser regions of the graph.

In terms of accuracy and recall, as depicted in Figures 15a and 15b, most datasets have minimal changes given different LID threshold.

In conclusion, tuning the LID threshold primarily enhances computational efficiency by optimizing the number of skips, thereby reducing inference time, without affecting accuracy and recall.

### 3.5 COMPLEXITY ANALYSIS

#### 3.5.1 SEARCH COMPLEXITY

The search complexity of HNSW++ builds upon the original HNSW framework, integrating dual-branch navigation and LID-driven optimizations. By employing a dual-branch structure, the search process is divided between two branches, each exploring distinct graph regions. Starting at the top layers of both branches, the algorithm ensures robust exploration and minimizes the risk of search stagnation in local minima.

Incorporating layer-skipping bridges, the complexity is further reduced by allowing direct transitions to lower layers when LID conditions are met. Let $P_{\text{skip}}$ represent the probability of a skip occurring based on a node's LID exceeding the threshold $T$, and $L_{\text{total}}$ represent the total number of layers. The expected number of layers traversed is reduced to $L_{\text{total}} \cdot (1 - P_{\text{skip}})$, making the effective layer exploration more efficient than the standard HNSW.

Each branch independently executes the search with complexity scaling as $\mathcal{O}(\log(N))$ due to the hierarchical structure. The merging of results at the base layer introduces a constant overhead, maintaining the overall search complexity at $\mathcal{O}(\log(N))$. Experimental results confirm that dual-branch navigation with selective layer skipping does not compromise logarithmic scaling, even in higher-dimensional datasets.

### 3.5.2 CONSTRUCTION COMPLEXITY

Construction in HNSW++ follows a layered insertion protocol, where nodes are assigned layers based on normalized LID values. The dual-branch structure halves the effective dataset size per branch, reducing the computational load for insertion operations. Each node is placed after executing nearest neighbor searches at each layer, with complexity per insertion scaling logarithmically as $\mathcal{O}(\log(N))$.

The layer-skipping mechanism also optimizes insertion by allowing high-LID nodes to directly establish connections in deeper layers, bypassing intermediate ones. With $M_{\max}$ as the maximum number of connections per node and $k$ as the neighbor set size, the average insertion time is proportional to $\mathcal{O}(M_{\max} \cdot \log(N))$ per node. Thus, the overall construction complexity for a dataset of size $N$ is $\mathcal{O}(N \cdot \log(N))$, consistent with the standard HNSW scaling.

Assuming the LID values are provided beforehand, HNSW++ does not add significant computational overhead. The LID-based assignment process scales efficiently, as the majority of the computational load is absorbed during initial LID calculations. Consequently, HNSW++ retains the scalability of the original HNSW construction process.

## 4 EXPERIMENTS

The experiments were conducted on a system equipped with an AMD EPYC 7542 32-Core Processor (64 threads, 2 threads per core) and 80 GB of RAM, running a 64-bit Debian OS. To ensure a fair comparison, the best performance results for each algorithm were chosen based on recall across varying thresholds. The HNSW++ code were implemented in C++ (and Python as extension in Appendix).

For ground truth evaluation in Python verrsion, we employed Scikit-learn's NearestNeighbors function run by $k$-Nearest-Neighbours ($k$NN) algorithm to generate accurate nearest neighbor comparisons.

Different hyperparameter sets are employed for each algorithm to ensure optimal performance and a fair comparison across all methods.

In this section, we provide (1) An overview of the datasets, (2) Main results - Performance comparison between the state-of-the-art approaches and HNSW++, (3) Ablation study - Performance comparison between original HNSW, LID-based HNSW, Multi-branch HNSW, and HNSW++.

### 4.1 DATASETS OVERVIEW

Our experiments leveraged six datasets spanning various domains, including Computer Vision (Jégou et al., 2011; Wolf et al., 2011; Babenko & Lempitsky, 2016), Natural Language Processing (Pennington et al., 2014), and randomly generated vectors. These datasets exhibited a range of Local Intrinsic Dimensionality (LID) values and dimensionalities. All distance computations were performed in $L_2$ space for consistency. Due to resource limitations, the experiments were conducted using 10,000 data points for graph construction and 1,000 data points for inference (Table 1).

The LID of each data point was computed using Maximum Likelihood Estimation (MLE), considering the exact nearest neighbors defined by ef_construction (128). The distribution of LID values for each dataset is visualized in Figure 18.

| Dataset | d | Space | Data points | LID Avg | LID Median | Type |
|---------|-----|-------|-------------|---------|------------|------|
| GLOVE | 100 | $L_2$ | 11,000 | 31.94 | 30.52 | NLP |
| SIFT | 128 | $L_2$ | 11,000 | 14.75 | 14.81 | CV |
| RANDOM | 100 | $L_2$ | 11,000 | 42.75 | 42.54 | Synthetic |
| DEEP | 96 | $L_2$ | 11,000 | 16.42 | 16.22 | CV |
| GIST | 960 | $L_2$ | 11,000 | 28.30 | 28.67 | CV |
| GAUSSIAN | 12 | $L_2$ | 11,000 | 22.60 | 12.80 | Synthetic |

Table 1: LID Averages, Medians, and Computation Times for Different Datasets.

## 4.2 EXPERIMENT RESULTS

To evaluate our algorithm, several methods were utilized for comparison. Since our code for HNSW++ was implemented in Python, and the other methods use different programming languages that offer faster runtimes than Python, we cannot fairly compare the construction time and inference time. Therefore, we focus on performance aspects only to ensure fairness. Further experiments including accuracy, recall, construction, and query time are conducted in Section 4.3. The algorithms we use to compare with HNSW++ are:

- FAISS IVFPQ (Inverted File Index with Product Quantization)[1] (Johnson & Jégou, 2019) : Combines inverted file indexing with product quantization to perform efficient approximate nearest neighbor searches on large-scale datasets.

- NMSLib (Non-Metric Space Library)[2]: A highly optimized library designed for approximate nearest neighbor search, built upon HNSW. :

- PyNNDescent[3]: Implements an efficient, approximate nearest neighbor search using Nearest Neighbor Descent graphs.

- Annoy (Approximate Nearest Neighbors Oh Yeah)[4]: Utilizes a forest of random projection trees to perform nearest neighbor searches. It follows a graph-based approach.

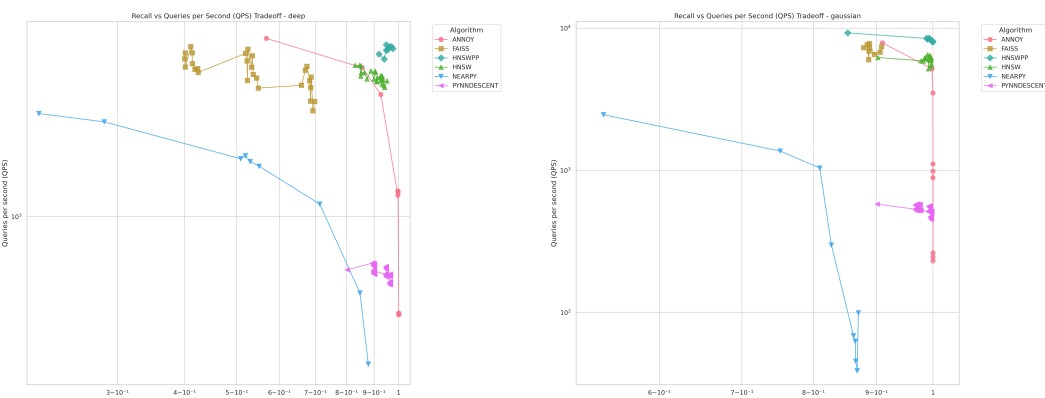

Figure 4: Illustration of recall of all algorithms on Deep dataset.

Figure 5: Illustration of recall of all algorithms on Gaussian dataset

The HNSW++ algorithm consistently outperforms its competitors across various datasets, particularly in high-dimensional and moderate-LID environments. For example, in datasets such as GIST (960 dimensions, LID Avg 28.30), SIFT (128 dimensions, LID Avg 14.75), GAUSSIAN (12 dimensions, LID Avg 22.60), RANDOM (100 dimensions, LÍ Avg 42.75) and DEEP (96 dimensions, LID Avg 16.42), HNSW++ demonstrates exceptional recall@10 in C++ (Figure 4, 5, 6, 7, 8) and accuracy@10, recall@10 in Python(Figure 16 and Figure 17, outperforming other methods due to its dual-branch architecture and advanced graph traversal techniques. These results illustrate HNSW++'s ability to effectively manage complex search spaces, avoiding local minima and optimizing search paths. HNSW++ maintains superior recall, accuracy in both C++ and Python compared to traditional methods such as FAISS (IVFPQ), NMSLIB (HNSW) and Annoy, making it one of the best-performing algorithms overall.

HNSW++ demonstrates exceptional performance in construction times, significantly surpassing both PyNNDescent and NMSLIB (HNSW) in efficiency (Figure 9). When compared to FAISS (IVFPQ), one of the leading state-of-the-art methods for approximate nearest neighbor search, HNSW++ exhibits only a marginal 9% difference in construction time. This comparison is based on a comprehensive evaluation conducted over 100 independent runs across all datasets, highlighting

---

[1]https://github.com/facebookresearch/faiss

[2]https://github.com/nmslib/nmslib

[3]https://github.com/lmcinnes/pynndescent

[4]https://github.com/spotify/annoy

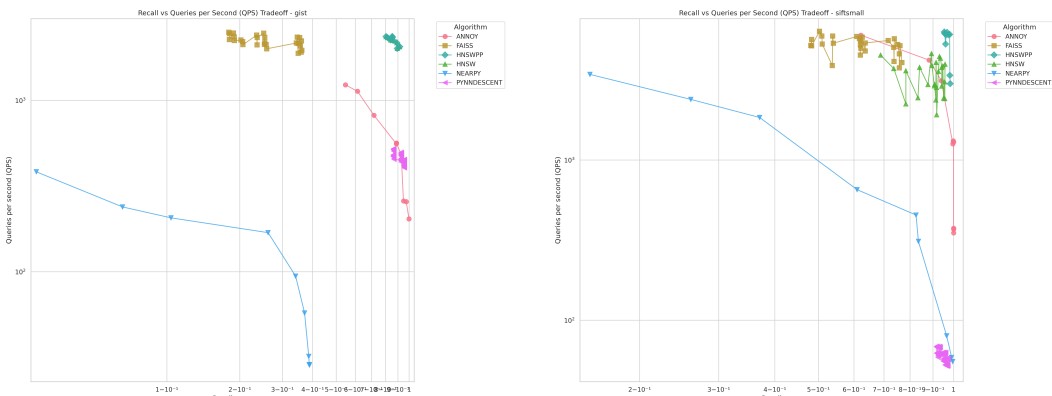

Figure 6: Illustration of recall of all algorithms on GIST dataset.

Figure 7: Illustration of recall of all algorithms on SIFT dataset

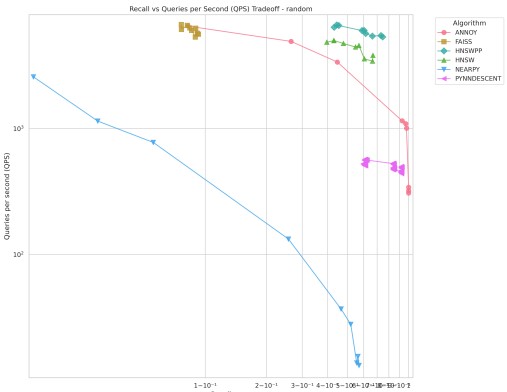

Figure 8: Illustration of recall of all algorithms on Random dataset.

the robustness and scalability of HNSW++ in diverse data environments. These results underscore the method's ability to balance speed and accuracy, making it a competitive choice for large-scale applications.

## 4.3 ABLATION STUDY

To gain a deeper understanding of the contribution of individual factors to recall, accuracy, construction time, and query time, we conducted a set of ablation experiments. Each experiment focused on a specific aspect of the algorithm by evaluating the following configurations:

- Basic: The standard HNSW algorithm without any modifications.

- Multi-Branch: This version retains only the two parallel branches of HNSW, excluding both the LID-based insertion mechanism and the skip-layer approach.

- LID-Based: This variant utilizes only the LID-based insertion mechanism, removing the two-branch structure and the skip-layer mechanism.

- HNSW++: The full version incorporating all three enhancements: two branches, the LID-based insertion mechanism, and the skip-layer mechanism.

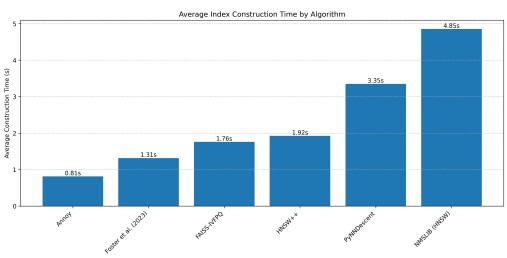

Figure 9: Illustration of recall of all algorithms on Random dataset.

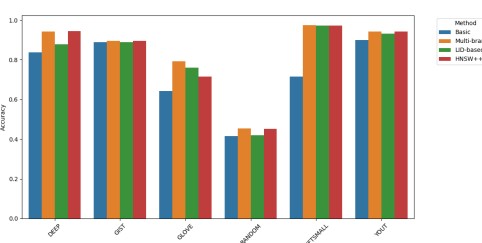

Figure 10: Illustration of the average accuracy for each algorithm in ablation study.

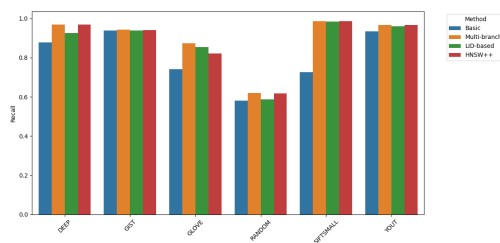

Figure 11: Illustration of the average recall for each algorithm in ablation study.

As shown in Figures 10 and 11, the Multi-Branch and HNSW++ algorithms consistently achieve the highest accuracy and recall across all five datasets, outperforming the other variants. However, in the Natural Language Processing dataset (GLOVE), the LID-Based algorithm surpasses HNSW++ in both accuracy and recall. In all tasks, the Basic algorithm consistently yields the lowest accuracy and recall scores. These results clearly demonstrate that the Multi-Branch and LID-Based methods significantly enhance the accuracy and recall of the Basic algorithm, which ultimately culminates in the combined strengths of HNSW++.

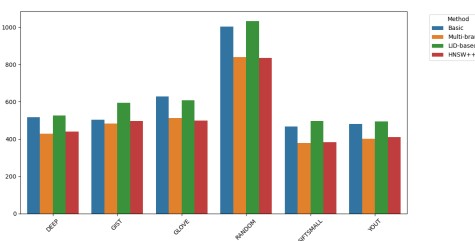

Figure 12: Illustration of the average construction time for each algorithm in ablation study.

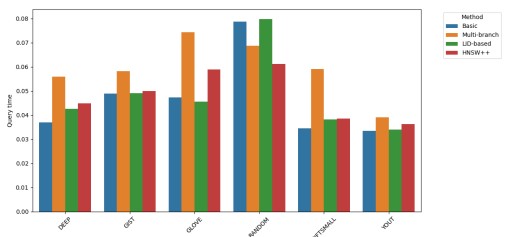

Figure 13: Illustration of the average query time for each algorithm in ablation study.

Regarding construction time, HNSW++ and Multi-Branch show similar performance, alternating as the fastest on different datasets. Both consistently outperform Basic by approximately 16-20%, and LID-Based by 18-22% across all six datasets.

In terms of query time, Multi-Branch tends to be slower than the other three algorithms in most tasks, with the exception of the random dataset. The query speeds of Basic, LID-Based, and HNSW++ are nearly identical, differing by only 1-2% across most tasks, except for the random dataset, where both Multi-Branch and LID-Based are noticeably slower than HNSW++.

In summary, HNSW++ stands out for its well-rounded performance, combining the strengths of the Multi-Branch and LID-Based mechanisms. Its construction time, in particular, represents a significant improvement over the Basic algorithm, highlighting its effectiveness in various tasks.

## 5  CONCLUSION

In this work, we introduce HNSW++, a novel algorithm that addresses several key challenges in nearest neighbor search. HNSW++ is designed to reduce the probability of falling into local minima, improve the detection of outlier nodes, and enhance the connectivity of clusters. It also significantly accelerates the construction process while maintaining stable inference time. To achieve these advancements, we integrate innovative mechanisms such as a multi-branch structure, insertion based on Local Intrinsic Dimensionality (LID) values, and a skip-layer approach, ensuring the algorithm's efficiency and scalability.

Furthermore, we perform an in-depth analysis of how each of these factors contributes to the overall performance of HNSW++, providing insights into their individual and combined effects. Through extensive experimentation on a wide range of tasks, datasets, and benchmarks, HNSW++ consistently achieves state-of-the-art performance, demonstrating both superior accuracy and faster execution times compared to existing methods. This highlights the potential of HNSW++ as a robust solution for large-scale nearest neighbor search problems.

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

# A    APPENDIX

## A.1    LID THRESHOLD INFLUENCE ON NUMBER OF SKIPS AND PERFORMANCE

The provided figures illustrate the results of experiments analyzing the effect of varying thresholds on different aspects of the HNSW algorithm's performance.

- Figure 14 shows the effect of different thresholds on the number of layers skipped during query searches. As the threshold increases, the number of layers bypassed by the search algorithms decreases significantly across all datasets.
- Figure 15a and Figure 15b depict the impact of threshold values on accuracy and recall across six datasets. The two figures show that showing that most algorithms maintain stable accuracy across different threshold values, with minimal variance as the threshold increases.

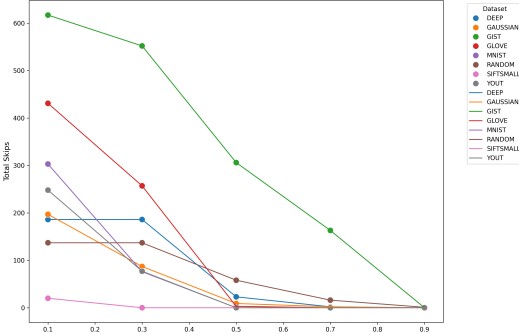

Figure 14: Illustration the effect of varying the threshold on the number of layers bypassed by each algorithm after 1000 query searches.

## A.2    PSEUDOCODES

### A.2.1    INSERT

The Insert algorithm constructs the hierarchical graph structure in HNSW++. Specifically, given the assigned layer and branch index for the new element $q$, the entry point $ep$ starts traversing from the top layer of that branch and stops at the designated layer. At this point, $q$ is added to the graph with edges connecting it to its neighbors, up to a maximum of $maxk$ neighbors. During the insertion process, a dynamic candidate list of size $efConstruction$ is maintained. The node $q$ is represented by its corresponding matrix.

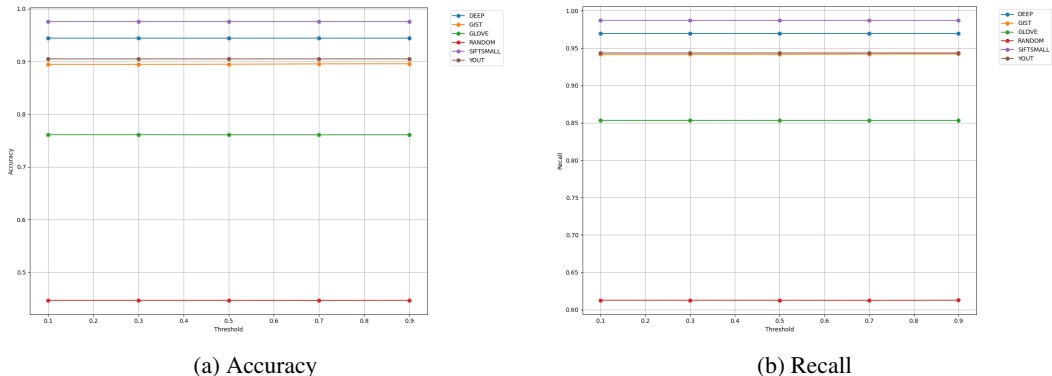

(a) Accuracy            (b) Recall

Figure 15: Illustration of impact of threshold on accuracy and recall of all algorithms across 6 datasets.

---

**Algorithm 1** INSERT

---

**Input:** $hnsw$ - multilayer HNSW graph structure, $q$ - new element (point) to be inserted, $assigned\_layers$ - mapping of element labels to assigned layers, $assigned\_branches$ - mapping of element labels to assigned branches, $branch0, branch1$ - two branches of the HNSW graph, $base\_layer$ - base layer of the HNSW graph **Output:** Update $hnsw$ by inserting element $q$

1: Retrieve $layer \leftarrow assigned\_layers[q.label]$
2: Retrieve $branch \leftarrow assigned\_branches[q.label]$
3: **if** $branch = 0$ **then**
4:    $branch0.setLevel(layer)$
5:    $branch0.setConnectState(layer \neq 0)$                   ▷ Set connection state: true for upper layers, false for base layer
6:    $branch0.addPoint(q, q.label)$
7:    $closest \leftarrow branch0.getClosestPoint()$
8: **else**
9:    $branch1.setLevel(layer)$
10:    $branch1.setConnectState(layer \neq 0)$
11:    $branch1.addPoint(q, q.label)$
12:    $closest \leftarrow branch1.getClosestPoint()$
13: **end if**
14: $base\_layer.setEnterpointNode(closest)$
15: $base\_layer.addPoint(q, q.label)$
16: **return** updated $hnsw$

---

### A.2.2 SEARCH

The Search algorithm is complex due to the integration of Multi-branch and Skipping methods. Initially, for layers greater than 0, the search identifies one nearest neighbor for each branch, setting it as the entry point $ep$ for the next layer of that branch. If a branch triggers a $skip$, the search for that branch will continue at layer 0, while it waits for the other branch to complete its search. If both branches reach layer 0 simultaneously, or if $branch_0$ reaches layer 0 first, $branch_0$ will initiate the search with $ef\_search$, resulting in $W_1$, the set of $k$ nearest neighbors from $branch_0$. When $branch_1$ starts its search, it will use $W_1$ as an $exclude\_set$ to avoid returning the same neighbors as $branch_0$ (Fig.19) . Conversely, if $branch_1$ reaches layer 0 first, the roles are reversed. Once the searches from both branches are completed, $W_1$ and $W_2$ are combined to retrieve the final set of $k$ nearest neighbors.

---

**Algorithm 2** SEARCH

**Input:** HNSW hierarchical graph, query element $q$, number of neighbors $efSearch$, final number of neighbors $k$, LID threshold $lid\_threshold$ (optional)

**Output:** Combined nearest neighbors $W$, number of skips $skip\_count$

1: Initialize $W_1 \leftarrow \{entrance1\}, W_2 \leftarrow \{entrance2\}$          ▷ Nearest neighbors for both branches
2: $layer_1 \leftarrow \text{TopLayer}(HNSW, 0), layer_2 \leftarrow \text{TopLayer}(HNSW, 1)$
3: $skip\_count \leftarrow 0$
4: **while** $layer_1 \geq 0$ **or** $layer_2 \geq 0$ **do**
5:     **if** $layer_1 \geq 0$ **then**
6:         $W_1, skip_1 \leftarrow \text{SEARCH-LAYER}(q, W_1, ef, layer_1, 0, lid\_threshold)$
7:         **if** $skip_1$ is **True then**
8:             $layer_1 \leftarrow 0$          ▷ Skip remaining layers for branch 1
9:             $skip\_count \leftarrow skip\_count + 1$
10:         **else**
11:             $layer_1 \leftarrow layer_1 - 1$
12:         **end if**
13:     **end if**
14:     **if** $layer_2 \geq 0$ **then**
15:         $W_2, skip_2 \leftarrow \text{SEARCH-LAYER}(q, W_2, ef, layer_2, 1, lid\_threshold)$
16:         **if** $skip_2$ is **True then**
17:             $layer_2 \leftarrow 0$          ▷ Skip remaining layers for branch 2
18:             $skip\_count \leftarrow skip\_count + 1$
19:         **else**
20:             $layer_2 \leftarrow layer_2 - 1$
21:         **end if**
22:     **end if**
23: **end while**
24: $W \leftarrow \text{Top-}k(W_1 \cup W_2)$          ▷ Combine and select top $k$ neighbors
25: **return** $W, skip\_count$

---

The Search Layer algorithm is designed to operate within the HNSW++ hierarchical graph structure. It takes as input the hierarchical graph $HNSW$, the current layer index $layer\_i$, the query element index $q$, the starting entry point $ep$, the number of neighbors to return $ef$, the branch index $world$, an optional LID threshold $lid\_threshold$, and an optional set of nodes to exclude $exclude\_set$.

Initially, the algorithm checks whether the entry point $ep$ is part of the $exclude\_set$. If so, it searches for a new entry point that is not in the exclude set by exploring neighboring nodes.

Next, the search begins by identifying the nearest neighbors for the query node $q$ within the specified layer. The entry point $ep$ is initialized as both the first candidate and the initial nearest neighbor. The algorithm iteratively selects the closest candidate $c$, compares its distance to $q$ with that of the furthest neighbor $f$ in the current nearest neighbors set, and updates the set if $c$ is closer. This process continues until the desired $ef$ nearest neighbors are identified, gradually refining the search by exploring neighboring nodes.

Before returning the final set of $ef$ neighbors, the algorithm performs an additional check. If the distance from the nearest node in the $ef$ neighbors set to the query point is less than the dataset's average distance, and the node's normalized LID value exceeds the threshold, the algorithm will return a skip signal. In this case, the neighbor set $W$ and the skip signal are passed to Algorithm 2 for further processing.

### A.2.3 ASSIGN LAYER

The Assign Layer algorithm is responsible for determining the layer placement of all nodes prior to graph construction. It begins by calculating the expected sizes of each layer and stores them in an array. Using the array of nodes sorted in descending order by their LID values, the algorithm assigns each node to a layer according to the corresponding expected layer sizes.

**Algorithm 4** ASSIGN_LAYER

**Input:** $topL$ - maximum number of layers, $mL$ - normalization factor for level generation, $normalized\_LIDs$ - array of normalized local intrinsic dimensionalities

**Output:** $assigned\_layers$ - an array of [layer, branch] assignments for each node

1: $n \leftarrow$ length of $normalized\_LIDs$
2: $branch\_0\_size \leftarrow \lceil n/2 \rceil$, $branch\_1\_size \leftarrow \lfloor n/2 \rfloor$
3: Initialize arrays $expected\_layer\_size$ for both branches with size $topL$
4: **for** each branch in $\{0, 1\}$ **do**
5:      **for** each node in branch **do**
6:          $layer\_i \leftarrow \max(\min(\lfloor -\log(\text{random}()) * mL \rfloor, topL - 1), 0)$
7:          Increment $expected\_layer\_size[branch][layer\_i]$
8:      **end for**
9: **end for**
10: Sort indices of $normalized\_LIDs$ in descending order
11: Initialize $assigned\_layers$ with shape $(n, 2)$ to hold [layer, branch]
12: Initialize $current\_layer\_size$ for both branches to zero
13: $current\_branch \leftarrow 0$                                            ▷ Start with branch 0
14: **for** each sorted index **do**
15:      $branch \leftarrow current\_branch$
16:      **for** $layer$ from $topL - 1$ down to 0 **do**
17:          **if** $current\_layer\_size[branch][layer] < expected\_layer\_size[branch][layer]$ **then**
18:              Assign node to layer and branch in $assigned\_layers$
19:              Increment $current\_layer\_size[branch][layer]$
20:              **Break**
21:          **end if**
22:      **end for**
23:      Alternate between branches (switch $current\_branch$)
24:      **If** one branch is full, assign the remaining nodes to the other branch
25: **end for**
26: **return** $assigned\_layers$

### A.2.4 NORMALIZE LIDs

Given an array of LID values, this function performs normalization using min-max normalization, as defined by the following equation:

$$\text{normalized\_LID}(x) = \frac{x - \min(\text{LID})}{\max(\text{LID}) - \min(\text{LID})} \tag{5}$$

This ensures that the LID values are scaled to a range between 0 and 1, facilitating consistent comparisons across nodes.

**Algorithm 5** NORMALIZE_LIDS

**Input:** $lids$ - array of local intrinsic dimensionalities
**Output:** $normalized\_LIDs$ - array of normalized LIDs

1: $min\_lid \leftarrow$ minimum of $lids$
2: $max\_lid \leftarrow$ maximum of $lids$
3: $normalized\_LIDs \leftarrow (lids - min\_lid)/(max\_lid - min\_lid)$
4: **return** $normalized\_LIDs$

### A.3 HNSW++ IN PYTHON

### A.4 LID DISTRIBUTION

### A.5 LID DISTRIBUTION

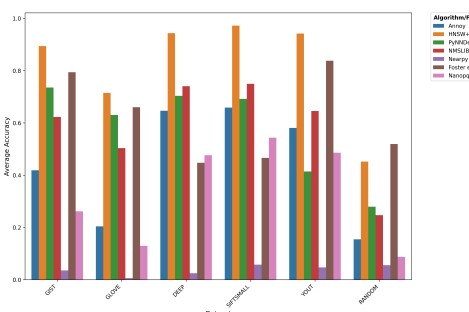

Figure 16: Illustration of accuracy of all algorithms across 6 datasets in Python.

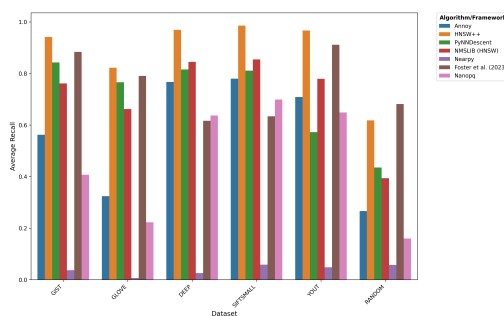

Figure 17: Illustration of recall of all algorithms across 6 datasets in Python.

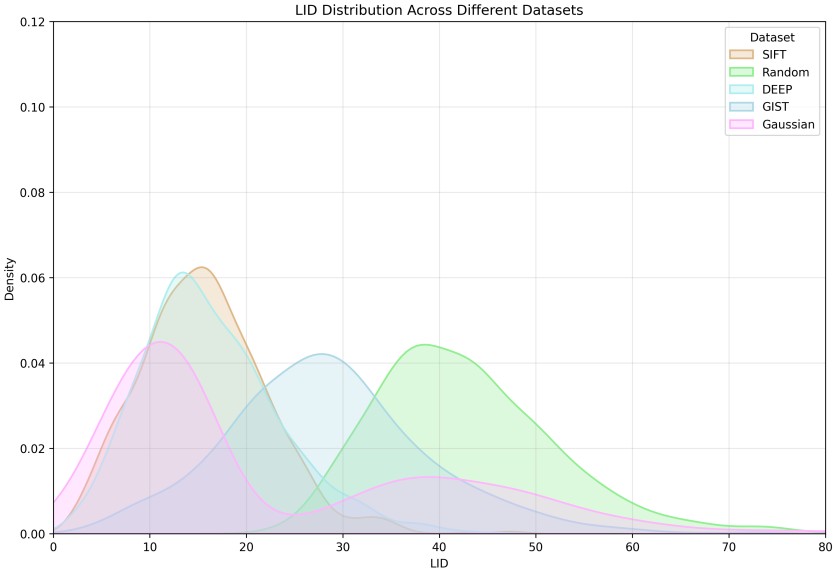

Figure 18: LID distribution across different datasets.

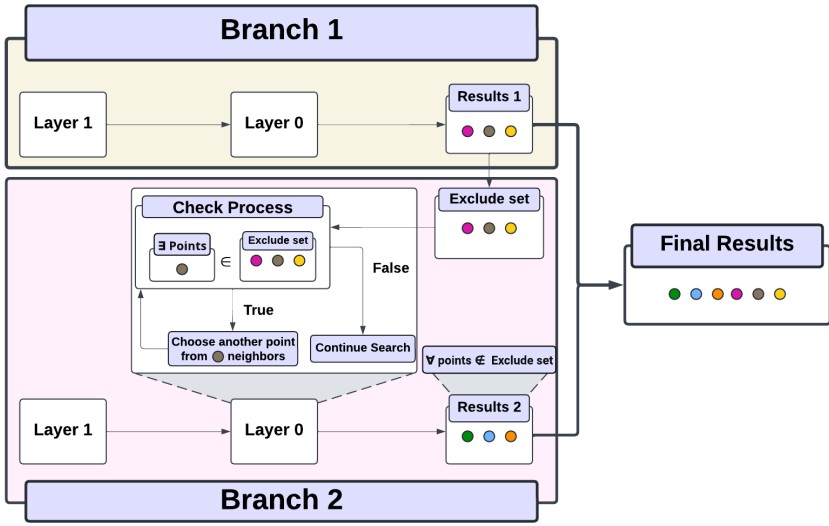

Figure 19: Illustration of Workflow of Exclude_set.