# OpenReview forum: "Dual-Branch HNSW Approach with Skip Bridges and LID-Driven Optimization"
_ICLR.cc/2025/Conference — Submitted to ICLR 2025_

### Official Review · Reviewer_NYMT · 2024-11-02

**Soundness:** 3
**Presentation:** 3
**Contribution:** 2
**Rating:** 5
**Confidence:** 4

**Summary:**

This paper presents an enhanced Hierarchical Navigable Small World (HNSW) algorithm addressing limitations in local optima and scalability by introducing a dual-branch structure with LID-based insertion and a bridge-building shortcut technique. These innovations improve cluster connectivity, capture outliers more effectively, and reduce inference time. Experimental results across NLP, DL, and CV datasets demonstrate notable accuracy and speed improvements over the original HNSW algorithm.

**Strengths:**

1)This paper presents a novel enhancement to the HNSW algorithm, addressing key limitations related to local optima and inference speed.
2)The research is evaluated across diverse benchmarks, including datasets from Computer Vision (CV), Deep Learning (DL), and Natural Language Processing (NLP). The experiments clearly support the proposed method's superiority in both accuracy and speed, with substantial performance gains.
3）The paper is well-written, with clear explanations of complex concepts and methods.

**Weaknesses:**

1)The related work section is overly concise, lacking a comprehensive review of current research, which limits contextual understanding of the contributions.
2)Although the paper claims improvements in inference speed, Figures 9 and 10 show only modest gains, casting doubt on the practical significance of this claim.
3)The experimental evaluations are relatively limited, with insufficient algorithmic comparisons; in particular, using only the GLOVE dataset for NLP benchmarks diminishes the persuasiveness of the results in this domain.

**Questions:**

The paper introduces an optimization algorithm for HNSW,  including pseudocode for the dual-branch structure, LID-based insertion, and bridge-building techniques would enhance clarity and understanding.

---

> ### Author Response · Authors · 2024-11-21
> **Response to Reviewer 3**
>
> We sincerely appreciate your thoughtful and constructive feedback. We would like to provide our detailed responses to each of your comments below:
>
> - The related work section is overly concise, lacking a comprehensive review of current research, which limits contextual understanding of the contributions
>
> __We greatly appreciate your insight and fully agree with this point. We will enhance our related work section to give a better overview of the current literature. The changes will be highlighted in the updated manuscript for your convenience to review.__
>
> - Although the paper claims improvements in inference speed, Figures 9 and 10 show only modest gains, casting doubt on the practical significance of this claim
>
> __Thank you very much for your feedback. As mentioned in our Abstract, we would like to clarify that we claimed an improvement only in construction speed, not inference speed. Our inference speed remains unchanged, which is why Figure 10 (inference speed comparison) shows modest gains. Regarding construction time, we have reduced it by up to 20% compared to the original HNSW algorithm.__
>
> __However, we recognize that our presentation may have caused some confusion, and we apologize for this oversight. We will revise the writing and diagrams in the updated manuscript to convey the message more effectively.__
>
> - The experimental evaluations are relatively limited, with insufficient algorithmic comparisons; in particular, using only the GLOVE dataset for NLP benchmarks diminishes the persuasiveness of the results in this domain
>
> __Thank you for pointing this out. We understand the concerns regarding the limited experimental evaluations. Based on your comments, as well as those from other reviewers, we have decided to add ANN Benchmark [3], which includes 13 datasets from widely recognized domains in nearest neighbor search. We selected datasets from diverse fields, including Natural Language Processing (NLP) and Computer Vision (e.g., GloVe, SIFT), to demonstrate the versatility of our method across different domains. Additionally, our evaluations adhere to the standards established in notable ANN papers by Peng [4], Malkov [5], and Wang [6]. Given this, we believe that conducting an exhaustive evaluation on specialized domain datasets may not be suitable at this stage.__
>
> __[3] Aumüller, M., Bernhardsson, E., & Faithfull, A. (2018). ANN-Benchmarks: A Benchmarking Tool for Approximate Nearest Neighbor Algorithms. Retrieved from https://ann-benchmarks.com/
> [4] Peng, Y., Choi, B., & Chan, T. N. (2023). Efficient Approximate Nearest Neighbor Search in Multi-dimensional Databases. Proceedings of the ACM SIGMOD International Conference on Management of Data, Seattle, WA, USA
> [5] Malkov, Y. A., & Yashunin, D. A. (2018). Efficient and robust approximate nearest neighbor search using Hierarchical Navigable Small World graphs.
> [6] Wang, M., Xu, X., Yue, Q., & Wang, Y. (2021). A comprehensive survey and experimental comparison of graph-based approximate nearest neighbor search__
>
> - The paper introduces an optimization algorithm for HNSW, including pseudocode for the dual-branch structure, LID-based insertion, and bridge-building techniques would enhance clarity and understanding
>
> __We did provide the pseudocode for dual-branch structure, LID-based insertion, and bridge-building techniques in the Appendix section. Please let us know if there are any improvements that should be made.__
>
> - Conclusion
>
> __We hope our responses provide some clarity and contribute positively to your evaluation process. We are committed to revising our approach and adding the above points to our updated manuscript. Thank you again for your valuable feedback.__

---

> ### Author Response · Authors · 2024-11-30
> **Revised manuscript submitted**
>
> Thank you for your constructive feedback.
>
> To align with your insightful feedback, we have made several updates:
> - We have enriched the "Related Work" section by incorporating additional context and highlighting state-of-the-art advancements from the most recent algorithms
> - We have refined the wording throughout the paper to more effectively convey the core message of our algorithm (Increase accuracy, Increase Recall, Improve Construction Time, Maintain Inference Time)
> - We have also updated the experimental section, leveraging C++ implementations and carefully selected datasets from the ANN benchmark. The results from the new C++ benchmark demonstrate the very promising performance of our algorithm (figures 5,6,7,8,9)
>
> We hope these updates can clarify your doubts and potentially improve our score. Please let us know if there is anything you are unsure about; we would be more than happy to provide further explanation.

---

### Official Review · Reviewer_7mVb · 2024-11-03

**Soundness:** 2
**Presentation:** 2
**Contribution:** 2
**Rating:** 5
**Confidence:** 3

**Summary:**

This paper The proposed HNSW++ algorithm, which introduces a dual-branch structure, LID-based node insertion, and skip-layer bridges to address the limitations of the original HNSW, such as local optima and cluster disconnections. Experiments have shown the that the proposed method is competitive both in performance and in inference speed.

**Strengths:**

1. The dual-branch structure and LID-based insertion mechanism are well-motivated and novel.

1. The experiments have (partially) shown the effectiveness of the proposed method.

**Weaknesses:**

My major concerns are about experiments.

1. The proposed method is implemented in Python, which is not a good programming language model for comparing speed. The authors may provide a theoretical time complexity analysis to complement their empirical results. This would allow a fairer comparison of the algorithm's efficiency across different implementations.

1. The baselines are relatively weak, I did not see the advanced methods (e.g., IVFPQ) used in faiss[1]. I'd like to have authors justify their choice of baselines and explain why stronger baselines were not included.

[1] Johnson, Jeff, Matthijs Douze, and Hervé Jégou. "Billion-scale similarity search with GPUs." IEEE Transactions on Big Data 7.3 (2019): 535-547.

**Questions:**

please refer to weaknesses

---

> ### Author Response · Authors · 2024-11-21
> **Response to Reviewer 2**
>
> We sincerely appreciate your thoughtful and constructive feedback. We would like to provide our detailed responses to each of your comments below:
>
> - The proposed method is implemented in Python, which is not a good programming language model for comparing speed. The authors may provide a theoretical time complexity analysis to complement their empirical results. This would allow a fairer comparison of the algorithm's efficiency across different implementations
>
> __We completely agree with your suggestion and will incorporate C++ code to provide a more objective basis for speed comparisons. Additionally, we will also provide a detailed analysis and explanation of our algorithm time complexity to support the empirical results.__
>
> - The baselines are relatively weak, I did not see the advanced methods (e.g., IVFPQ) used in faiss[1]. I'd like to have authors justify their choice of baselines and explain why stronger baselines were not included
>
> __Thank you for your feedback. Implementing the algorithm in Python has indeed limited our options for selecting baselines and has constrained us from leveraging stronger alternatives. To address this, we will transition our algorithm to C++ and adopt more robust baselines, such as IVFPQ utilized in FAISS, among other SOTA methods.__
>
> - Conclusion
>
> __Thank you very much for your feedback. We are committed to revising our approach and adding the above points to our updated manuscript.__

---

> ### Author Response · Authors · 2024-11-30
> **Revised manuscript submitted**
>
> Thank you for your constructive feedback.
>
> Following your recommendations, we have conducted additional experiments using C++ to be more suitable for comparing speed. Our method shows very good results on all metrics (accuracy, recall, construction time, and inference time - Figures 5,6,7,8,9).
>
> We have also included a detailed analysis of time complexity starting from line 308.
>
> Also following your feedback, we have incorporated FAISS-IVFPQ and cited [1] as a strong state-of-the-art baseline, which helps to clarify the contributions of our algorithm.
>
> [1]  Johnson, Jeff, Matthijs Douze, and Hervé Jégou. "Billion-scale similarity search with GPUs." IEEE Transactions on Big Data 7.3 (2019): 535-547.
>
> We hope these updates can clarify your doubts and potentially improve our score. Please let us know if there is anything you are unsure about; we would be more than happy to provide further explanation.

---

### Official Review · Reviewer_bWVH · 2024-11-03

**Soundness:** 1
**Presentation:** 1
**Contribution:** 1
**Rating:** 1
**Confidence:** 5

**Summary:**

This paper proposed a branching scheme to accelerate HSNW search. Supposedly, this scheme could find out some more useful starting point in the layer-0 of the HSNW method. The experimental setup is completely wrong and thus I can't draw a conclusion that experiments validate the claim.

**Strengths:**

- Work on an important problem for practical application.

**Weaknesses:**

- I implemented HSNW from scratch and published HSNW-related algorithm in top data mining conferences before. The experimental setup is completely nonsense to me. It should sort of follow-up the ANN-benchmark setup and that's a more reasonable way to show results.

- Apparently, the algorithm doesn't compare to the real HSNW implementation in wall clock time, and only compare their own variations.

- The algorithm seems to be implemented in Python only, which is sort of contradicting to the point of AKNN. Most HSNW algorithm is implemented in C, and there is a reason for that. Many algorithm improvement can't really benefit the search as the real-world hardware doesn't support well for the operations; or the asymptotic analysis doesn't align well with the real operation cost. Unless the authors showed their modified algorithm can accelerate in C (doubtful as there are so many branching and that possibly will cause memory read busy), it's not very convincing.

- It reads to me that the method is only adding more operations in top layers. So it has to at least numerically show how many IP/distance calculation it saves for the layer-0. Otherwise, the computational cost can only go up......

- So even under the query time reported in Fig. 10, it's not very promising. Just based on the Figure 10, I will say it's not a really working method.

**Questions:**

- The paper is not very straightforward to understand as it lacks a running example. I really don't know what exactly "exclude_set" contains. A running example or any pictorial example in Figure 3 helps.

The definition of LID (x) is unclear. LID(x) reads like query dependent, or say LID(q) is something we want. That means there will be a property per graph/query pair, but in Figure 2/3, there seem to be multiple high LID nodes. I'm not sure what exactly LID means.

---

> ### Author Response · Authors · 2024-11-21
> **Response to Reviewer 1**
>
> Thank you for your insightful comments. We appreciate your feedback and have responded to each point below.
> - It should sort of follow-up the ANN-benchmark
>
> __We appreciate the suggestion to follow the ANN-benchmark setup as it is a fairer approach to compare HNSW++ with other ANN algorithms. We will add this point to the updated version.__
>
> - The algorithm doesn't compare to the real HSNW in wall clock time, and only compare their own variations.
>
> __Thank you for your feedback. We would like to clarify that we did compare our HNSW++ with the original HNSW in wall clock time. In Section 4.3 (Ablation Study), we have Figures 7, 8, 9, and 10 comparing accuracy, recall, construction time, and query time between different versions of HNSW++ and the original HNSW. To ensure a fair comparison, we re-implemented the original HNSW in Python, strictly adhering to its pseudocode before comparing it with different versions of HNSW++.__
>
> - The algorithm seems to be implemented in Python only Most HSNW algorithm is implemented in C, and there is a reason for that.
>
> __Thank you for the very insightful feedback. We agree that implementing in C++ can support better comparison of our approach with others. We will add the experimental results with the new implementation in the updated manuscript. During our literature review, we observed that the original HNSW and all related libraries are written in C++. Just to confirm, you mean C++, not C, right?__
>
> - The method only adds more operations in top layers. So it has to at least show how many IP/distance calculations it saves for layer-0.
>
> __We very much appreciate your suggestion to visualize the number of IP/distance calculations saved for layer 0 and will provide this in our updated manuscript.__
>
> __Regarding the additional computational cost on the top layers, assuming the LID values are provided beforehand, HNSW++ does not add significant computational cost. Instead of using a log formula for layer construction, the algorithm receives an array of LID values representing the density of each point’s surroundings to determine layer construction (Algorithms 4 and 5). During the inference phase, layer 0 does not add performance impactful computational cost because each branch only searches for (ef_search/2) nodes.__
>
> - The query time reported in Fig. 10, it's not very promising.
>
> __Regarding the Figure 10, we would like to clarify that it is aligned with our intentions, as we claimed only an improvement in construction speed (not inference/query speed), as stated in our Abstract. The skip mechanism is designed to offset the extra query time caused by the multi-branch approach, where sequential inference of 2 branches increases query time. By optimizing query efficiency and reducing redundant computations, the skip mechanism improves HNSW++ query performance, as shown in Figure 10.__
>
> - I really don't know what "exclude_set" contains. A running example in Figure 3 helps.
>
> __We appreciate your suggestion for visualizing how the exclude_set works. To explain the concept of "exclude_set", with two concurrent entry points at layer 0, the search paths often produce overlapping results. The exclude_set ensures one path searches first and passes its results to the other path, reducing redundancy. We explained this in Section 3.2.__
>
> - The definition of LID (x) is unclear.
>
> __Thank you very much for pointing it out. We will ensure that the definition of LID(q) is clearer in the updated manuscript. Basically, the definition of LID(q) indicates its role in capturing the density of data points around the query, serving as a measure of local intrinsic dimensionality. LID values guide layer assignment, allowing HNSW++ to balance branching structures by leveraging density estimates. For a more detailed explanation of the LID formula, we have included [1] for your reference.__
>
> __Regarding the multiple high LID nodes in Figures 2 and 3, our HNSW++ is built on the principle of placing high LID nodes in higher layers [2] to enhance the assign_layer algorithm. Instead of assigning layers randomly, we now base the assignment on LID values. Specifically, as described in Algorithm 4, nodes with higher LID values are strategically placed in higher layers, optimizing the overall graph structure.__
>
> __[1] Elizaveta Levina and Peter J. Bickel. Maximum likelihood estimation of intrinsic dimension. In
> Proceedings of the 17th International Conference on Neural Information Processing Systems,
> NIPS’04, pp. 777–784, Cambridge, MA, USA, 2004. MIT Press.
> [2] O. P. Elliott and J. Clark. The impacts of data, ordering, and intrinsic dimensionality on recall in hierarchical navigable small worlds. In Proceedings of the 2024 ACM Conference on Recommender Systems, 2024. doi: 10.1145/3664190.3672512.__
>
> - Conclusion
>
> __Thank you again for your constructive feedback. We are committed to revising our approach to demonstrate clearer practical improvements and better theoretical support in the updated manuscript.__

---

> > ### Comment · Reviewer_bWVH · 2024-11-23
> >
> > On LID definition.
> >
> > So I think your reply says LID is a query dependent metric. Now, if I change a query, you high LID nodes will change, and now your graph needs to be reorganized. That means your method will provide a dynamic upper level search graph for HSNW?
> >
> > If my understanding above is correct, do you include the re-format time of graph into wall clock time. If you did, what's the ratio of it among all the time you reported.

---

> ### Author Response · Authors · 2024-11-27
> **Response to Reviewer 1 (cont)**
>
> **1.So I think your reply says LID is a query dependent metric. Now, if I change a query, you high LID nodes will change, and now your graph needs to be reorganized. That means your method will provide a dynamic upper level search graph for HSNW?**
>
> We greatly appreciate your prompt feedback. To answer your question, let us first divide the algorithm into 2 distinct phases: 1) Construction phase and 2) Query phase. The LID values are exclusively utilized during the Construction phase (not in Query phase), where they determine the appropriate layer assignment for each point, as outlined in Algorithm 4. Therefore, the algorithm does **not** require reorganization for each query, since the LID values remain constant after the construction phase.
>
> **2. If my understanding above is correct, do you include the re-format time of graph into wall clock time. If you did, what's the ratio of it among all the time you reported.**
>
> As a result, there will be no need to account for the time or time ratio associated with algorithm reorganization.
>
> **3. Conclusion**
>
> We sincerely thank you for your invaluable and prompt feedback. Please let us know if you are still uncertain about any parts. We are more than happy to clarify further!

---

> > ### Comment · Reviewer_bWVH · 2024-11-30
> > **Conflict between replies**
> >
> > I believe the key confusion is that you can't know where the query lies in advance.
> >
> > So the definition of LID depends on the query, and it's only used in the construction phase, where you don't have access to query location/information. A contradiction.

---

> ### Author Response · Authors · 2024-11-30
> **Revised manuscript submitted**
>
> Thank you for your constructive feedback.
>
> We have converted our code to C++ and followed the ANN benchmark, selecting datasets suitable for high-dimensional and moderate-LID environments to ensure a fair comparison of our algorithm under wall-clock time conditions. We have very good results on all metrics (accuracy, recall, construction time, and inference time - Figures 5,6,7,8,9)
>
> We have also illustrated the role of the Exclude_set (Figure 19) and elaborated on the concept of LID in Section 2.2 of Related Works.
>
> We hope these updates can clarify your doubts and potentially improve our score. Please let us know if there is anything you are unsure about; we would be more than happy to provide further explanation.

---

> ### Author Response · Authors · 2024-12-01
>
> Thank you for your prompt reply. We can confirm that LID **does not** depend on the query. $LID(q)$ estimates the density of data points around any specific point $q$ in the dataset (not the query).
>
> To clarify:
>
> Let $\beta$ represent the LID value of any point $A$ in a dataset of $N$ points. $\beta(A)$ is defined as:
>
> $\beta(A) = -\left( \frac{1}{k} \sum_{i=1}^{k} \log \frac{||A - P_i||}{||A - P_k||} \right)$
>
> where $P_i$ represents the $i-th$ nearest neighbor of $A$, and $k$ is the number of neighbors considered.
>
> As we can see from the equation, LID usage has **nothing to do** with the query.
>
> Points with larger $\beta$ values (sparse or peripheral points) are typically placed in the upper layers of our HNSW++ graph. This placement enables the algorithm to:
> - Efficiently traverse sparse regions and transition between clusters during the top-down search process.
> - Focus on cluster-level movement in upper layers, reducing the chance of becoming "stuck" in dense clusters far from the query.
> - Progressively refine its search within denser regions in the lower layers, allowing it to converge toward the query's location.

---

### Meta-Review · Area_Chair_UW4W · 2024-12-19

**Metareview:**

The paper proposes HNSW++, an improved version of the Hierarchical Navigable Small World (HNSW) algorithm for approximate nearest neighbor (ANN) search. The dual-branch structure and LID-based insertion are novel contributions to the ANN search field. The reviewers argue that the initial submission had major flaws in its experimental setup. The explanation of how LID is calculated and used was initially unclear. The comparison to other state-of-the-art ANN algorithms was initially weak. The authors tried to add experiments to answer the reviewers' questions. The authors initially reported improved accuracy and speed across various datasets. However, a key reviewer (bWVH) heavily criticized the experimental setup for being flawed, particularly the use of Python (which is slower than C++ for this type of algorithm) and inconsistent benchmarking against standard ANN algorithms. After significant back-and-forth, the authors re-implemented in C++, used a standard benchmark (ANN-Benchmarks), and addressed several other criticisms regarding LID definition and experimental design. The authors responded positively and proactively to reviewer feedback. However, the reviewers' concerns were not fully addressed. I believe that the submission can be greatly improved by taking all the comments into consideration.

**Additional Comments On Reviewer Discussion:**

The authors initially reported improved accuracy and speed across various datasets. However, a key reviewer (bWVH) heavily criticized the experimental setup for being flawed, particularly the use of Python (which is slower than C++ for this type of algorithm) and inconsistent benchmarking against standard ANN algorithms. After significant back-and-forth, the authors re-implemented in C++, used a standard benchmark (ANN-Benchmarks), and addressed several other criticisms regarding LID definition and experimental design. The authors responded positively and proactively to reviewer feedback.

---

### Decision · Program_Chairs · 2025-01-22

Reject